# Determinant of zinc deficiency in orthopaedic inpatients

**Daisuke Iida, Tomonori Shigemura** *, **Yohei Yamamoto, Miki Morimoto, Yasuaki Murata**

Department of Orthopaedic Surgery, Teikyo University Chiba Medical Center, Ichihara, Chiba, Japan

* tshigepon@yahoo.co.jp

## Abstract

Zinc is vital for over 300 enzymes in major metabolic pathways, and deficiency can lead to serious conditions, especially post-surgery. This study aimed to investigate predictive factors of zinc deficiency in orthopaedic inpatients. A retrospective case-control study was conducted on patients admitted to Teikyo University Chiba Medical Center from 15 February to 31 August 2022. Patients were divided into zinc deficiency (< 60 µg/dL) and non-deficiency groups. Data included demographics, comorbidities, hospitalisation reasons, fracture details, medication use, and laboratory values. Fisher's exact test and two-sample t-tests were used for analysis. Of 156 patients, 47 (30.1%) had zinc deficiency. The case group had higher fracture rates (68.1% vs. 33.9%; p < 0.001), and lower rates of spinal disease (2.1% vs. 31.2%; p < 0.001) and osteoarthritis (8.5% vs. 22.9%; p = 0.04). Fragility and hip fractures were more common in the case group. Anaemia, hip fracture, and hypoalbuminaemia were independent predictive factors of zinc deficiency.

## Introduction

Zinc is a nutritionally essential trace mineral required for the activity of more than 300 enzymes involved in major metabolic pathways [1]. It is distributed throughout the body, with 85% located in muscles and bones, 11% in the skin and liver, and the remainder in other tissues [1]. A zinc deficiency is known to cause a variety of symptoms, including skin disorders [2], taste disorders [3], gonadal dysfunction in men [4], loss of appetite [5], diarrhoea [6,7], delayed wound healing [8,9], and increased susceptibility to infection [10].

These symptoms can lead to serious complications, particularly postoperatively. For instance, a prospective 15-month study of 80 consecutive patients undergoing total hip arthroplasty (THA) demonstrated an association between zinc deficiency and delayed wound healing in THA patients [8]. Another prospective study of 97 patients undergoing hip hemiarthroplasty for hip fractures concluded that serum zinc levels predicted delayed wound healing [9]. Delayed wound healing after joint replacement surgery increases the risk of prosthetic joint infection (PJI) [11,12]. PJI is one of the

**Data availability statement:** All relevant data are within the paper and Supporting Information files.

**Funding:** The author(s) received no specific funding for this work.

**Competing interests:** The authors have declared that no competing interests exist.

most serious complications after arthroplasties, not only reducing patients' quality of life [13], but also leading to longer hospital stays [14] and increased healthcare costs [15]. A study using data from the Nationwide Inpatient Sample in the United States demonstrated that length of stay was significantly longer for infected hip (9.7 days) and knee (7.6 days) arthroplasties compared to uninfected procedures (hip, 4.3 days; knee, 3.9 days) (p < 0.0001). The study also showed that hospitalisation charges were significantly greater for infected arthroplasties than for uninfected arthroplasties (hips, 1.76 times; knees, 1.52 times) (p < 0.0001) [16].

Understanding predictive factors of zinc deficiency before surgery is valuable for preventing symptoms arising from it. Furthermore, zinc deficiency has been linked to conditions such as cirrhosis [17,18], diabetes [19], chronic inflammatory bowel disease [20], and chronic kidney disease [21]. However, its association with orthopaedic disorders remains unclear. Additionally, predictive factors of zinc deficiency in orthopaedic inpatients before surgery are not well known.

To address these concerns, we conducted a retrospective case-control study, including patients with zinc deficiency in the case group and those without zinc deficiency in the control group, to examine predictive factors of zinc deficiency in hospitalised patients.

## Materials and methods

### Patient selection

The study included patients admitted to the Department of Orthopaedic Surgery of Teikyo University Chiba Medical Center between 15 February 2022 and 31 August 2022. The exclusion criteria were as follows: (1) patients whose zinc level was not measured during routine tests on admission, (2) patients who had previously been diagnosed with zinc deficiency, and (3) patients who were receiving zinc supplementation.

### Study design

The study used a case-control design, with patients with zinc deficiency included in the case group and those without zinc deficiency included in the control group. In this study, serum zinc concentration served as the primary marker for assessing zinc deficiency, given its widespread use and availability as an indicator of zinc deficiency [22–25]. Zinc deficiency was characterized by a serum zinc concentration of less than 60 μg/dL, in accordance with the treatment guidelines for zinc deficiency as published by the Japanese Society of Clinical Nutrition [26]. Serum samples were obtained at the time of admission. This study was approved by the Institutional Review Board of Teikyo University, IRB No.23-155). Because this was a retrospective study, the requirement for informed consent was waived.

### Data collection

The data used in this study were retrieved from the computerised database of Teikyo University Chiba Medical Center. The following data were retrieved from the patient

records: demographics (gender, age, height, weight, body mass index [BMI]), comorbidity (diabetes mellitus, hypothyroidism, gastroesophageal reflux disease, depression, haemodialysis), the reason for hospitalisation, details of fractures, medication use (zinc, iron, magnesium, Vitamin D, and calcium supplements, proton pump inhibitors, H2 receptor antagonists, diuretics, antihypertensive, opioid, and antibiotics), and laboratory data (zinc, calcium, albumin, creatinine, estimated glomerular filtration rate, haemoglobin, haemoglobin A1c, C-reactive protein [CRP], and alkaline phosphatase). Height and weight were measured using height scales (Okada medical supply Co., Ltd.) and weight scales (TANITA Corporation), respectively.

## Statistics

Means with standard deviations (SD) and percentages were used to report continuous and categorical variables, respectively. Statistical analyses were performed using Fisher's exact test for categorical variables and a two-sample t-test for continuous variables. To identify factors associated with zinc deficiency, multivariate logistic regression analysis (backward stepwise regression model) was performed for items that were significantly different from the univariate analysis. All p values were two-sided, and p values less than 0.05 were considered statistically significant. All statistical analyses were performed using EZR version 1.52 [27], which is a graphical user interface for R version 4.02 (The R Foundation for Statistical Computing, Vienna, Austria). Specifically, it is a modified version of the R commander designed to add statistical functions frequently used in biostatistics.

## Results

### Patient characteristics

Between 15 February 2022 and 31 August 2022, 201 patients were admitted to our hospital. Among these patients, 13 were excluded due to duplicate cases admitted during the study period, and 32 were excluded owing to incomplete data on serum zinc or other factors. None were excluded for having a prior diagnosis of zinc deficiency or for taking zinc supplements. The study ultimately included 156 patients, with a mean age of 68.1 years (SD: 16.8). This cohort comprised 67 men and 89 women. While there were no significant differences in age or BMI between the genders, notable differences were observed in height and weight. The men were significantly taller (1.66 m [0.07] vs. 1.51 m [0.07]; p < 0.001) and heavier (66.5 kg [16.3] vs. 52.2 kg [12.3]; p < 0.001) than the women. Regarding hospital admission reasons, the incidence of fractures showed no significant difference (p = 0.87). However, spinal diseases were more prevalent in men than in women (31.3% vs. 15.7%; p = 0.03), and osteoarthritis (OA) was less frequent in men (9.0% vs. 25.8%; p < 0.01). The fracture details between the two groups revealed no significant differences (Table 1).

We found that 47 (30.1%) patients had a zinc deficiency. The case and control groups were similar in terms of sex, height, and comorbidities, although there were significant differences in age, weight, and BMI. In particular, the mean (SD) age of the case group was significantly higher than that of the control group (75.5 years [13.8] vs. 65.0 years [17.1]; p < 0.001). The mean (SD) weight of the case group was significantly lower than that of the control group (53.8 kg [14.4] vs. 60.3 kg [16.1], p = 0.02). The mean (SD) BMI of the case group was significantly lower than that of the control group (21.7 kg/m$^2$ [4.5] vs. 24.0 kg/m$^2$ [5.2]; p < 0.01).

Regarding the reasons for hospitalisation, significant differences were observed between the case and control groups. The fracture rate was significantly higher in the case group than in the control group (68.1% vs. 33.9%; p < 0.001); spinal disease was significantly less common in the case group than in the control group (2.1% vs. 31.2%; p < 0.001); and OA was significantly less common in the case group than in the control group (8.5% vs. 22.9%; p = 0.04). The rates of fragility and hip and pelvic fractures were significantly higher in the case group than in the control group (p < 0.001, p < 0.001, p < 0.01, respectively). Among the medications, patients were significantly more likely to take iron supplements than controls (10.6% vs. 1.8%; p = 0.03) (Table 2).

**Table 1. Baseline characteristics of study patients.**

| Variables | Total (n = 156) | Men (n = 67) | Women (n = 89) | p value |
|---|---|---|---|---|
| Age, years | 68.1 ± 16.8 | 65.2 ± 16.5 | 70.4 ± 16.9 | 0.06 |
| Height, m | 1.58 ± 0.10 | 1.66 ± 0.07 | 1.51 ± 0.07 | < 0.001 * |
| Weight, kg | 58.4 ± 15.8 | 66.5 ± 16.3 | 52.2 ± 12.3 | < 0.001 * |
| BMI, kg/m² | 23.3 ± 5.1 | 24.0 ± 5.2 | 22.8 ± 5.0 | 0.12 |
| **Reason for hospitalisation** | | | | |
| Fracture | 69 (44.2%) | 29 (43.3%) | 40 (44.9%) | 0.87 |
| Spinal disease | 35 (22.4%) | 21 (31.3%) | 14 (15.7%) | 0.03 * |
| Osteoarthritis | 29 (18.6%) | 6 (9.0%) | 23 (25.8%) | < 0.01 * |
| Others | 23 (14.7%) | 11 (16.4%) | 12 (13.5%) | 0.65 |
| **Details of fractures** | | | | |
| Fragility fracture | 41 (26.3%) | 15 (22.4%) | 26 (29.2%) | 0.36 |
| Hip fracture | 28 (17.9%) | 15 (22.4%) | 13 (14.6%) | 0.29 |
| Proximal humeral fracture | 4 (2.6%) | 0 (0%) | 4 (4.5%) | 0.14 |
| Pelvic fracture | 6 (3.8%) | 3 (4.5%) | 3 (3.4%) | 1 |
| Distal radius fracture | 4 (2.6%) | 1 (1.5%) | 3 (3.4%) | 0.64 |
| Spinal fracture | 11 (7.1%) | 5 (7.5%) | 6 (6.7%) | 1 |

* Statistically significant.

BMI, body mass index.

## Laboratory findings

The case group exhibited a significantly lower mean (SD) serum zinc concentration of 46.2 μg/dL (9.7), compared to the control group's mean (SD) of 76.3 μg/dL (16.5) ($p < 0.001$). Additionally, patients with zinc deficiency had significantly lower mean (SD) blood concentrations of albumin (3.3 g/dL [0.7] vs. 4.0 g/dL [0.5]; $p < 0.001$) and haemoglobin (10.8 g/dL [2.0] vs. 13.1 g/dL [1.9]; $p < 0.001$) than the control group. Moreover, patients with zinc deficiency had significantly higher mean (SD) blood concentrations of CRP (2.9 mg/dL [4.2] vs. 1.2 mg/dL [2.6]; $p < 0.01$) than the control group.

The incidence of hypoalbuminaemia was significantly higher in the case group than in the control group (74.5% vs. 23.9%, $p < 0.001$). Finally, the incidence of anaemia was significantly higher in the case group than in the control group (70.2% vs. 16.5%; $p < 0.001$) (Table 3).

## Predictive factors of zinc deficiency

According to univariate analysis, statistically significant indicators ($p < 0.05$) were included in multivariate logistic regression analysis, including age ≥ 60 ($p < 0.01$), age ≥ 70 ($p < 0.01$), age ≥ 80 ($p < 0.01$), weight ($p = 0.02$), high BMI ($p = 0.01$), fracture ($p < 0.001$), spinal disease ($p < 0.001$), OA ($p = 0.04$), fragility fracture ($p < 0.001$), hip fracture ($p < 0.001$), pelvic fracture ($p < 0.01$), iron supplements ($p = 0.03$), hypoalbuminaemia ($p < 0.001$), anaemia ($p < 0.001$), and CRP ($p < 0.01$). The following indicators, which were not significant but had $p < 0.15$, were also included in multivariate logistic regression analysis, including age ≥ 50 ($p = 0.10$), age ≥ 90 ($p = 0.07$), low BMI ($p = 0.11$), H2 receptor antagonists ($p = 0.11$), and antihypertensive ($p = 0.11$). The multiple logistic regression analysis identified that anaemia (OR, 9.36; 95% CI, 3.35–26.10; $p < 0.001$), hip fracture (OR, 6.24; 95% CI, 1.96–19.90; $p < 0.01$), and hypoalbuminaemia (OR, 3.63; 95% CI, 1.33–9.89; $p < 0.05$) were independent predictive factors of zinc deficiency. The results also indicated that spinal disease was associated with a low predictive value of zinc deficiency (OR, 0.056; 95% CI, 0.006–0.49; $p < 0.01$) (Table 4).

**Table 2. Characteristics of zinc deficiency and controls.**

| Variables | Zinc deficiency (n = 47) | Control group (n = 109) | p value |
|---|---|---|---|
| **Demographics** | | | |
| Age, years | 75.5 ± 13.8 | 65.0 ± 17.1 | < 0.001 * |
| Age category | | | |
| 50 and above | 45 (95.7%) | 93 (85.3%) | 0.10 |
| 60 and above | 43 (91.5%) | 76 (69.7%) | < 0.01 * |
| 70 and above | 35 (74.5%) | 52 (47.7%) | < 0.01 * |
| 80 and above | 22 (46.8%) | 17 (15.6%) | < 0.01 * |
| 90 and above | 4 (8.5%) | 2 (1.8%) | 0.07 |
| Women | 25 (53.2%) | 64 (58.7%) | 0.60 |
| Height, m | 1.57 ± 0.10 | 1.58 ± 0.10 | 0.63 |
| Weight, kg | 53.8 ± 14.4 | 60.3 ± 16.1 | 0.02 * |
| BMI, kg/m$^2$ | 21.7 ± 4.5 | 24.0 ± 5.2 | < 0.01 * |
| High BMI ** | 8 (17.0%) | 40 (36.7%) | 0.01 * |
| Normal BMI ** | 27 (57.4%) | 54 (49.5%) | 0.39 |
| Low BMI ** | 12 (25.5%) | 15 (13.8%) | 0.11 |
| **Associated comorbidity** | | | |
| Diabetes mellitus | 11 (23.4%) | 24 (22.0%) | 0.84 |
| Hypothyroidism | 4 (8.5%) | 3 (2.8%) | 0.20 |
| GERD | 5 (10.6%) | 8 (7.3%) | 0.53 |
| Depression | 1 (2.1%) | 10 (9.2%) | 0.18 |
| Haemodialysis | 2 (4.3%) | 9 (8.3%) | 0.51 |
| **Reason for hospitalisation** | | | |
| Fracture | 32 (68.1%) | 37 (33.9%) | < 0.001 * |
| Spinal disease | 1 (2.1%) | 34 (31.2%) | < 0.001 * |
| Osteoarthritis | 4 (8.5%) | 25 (22.9%) | 0.04 * |
| Others | 10 (21.3%) | 13 (11.9%) | 0.15 |
| **Details of fractures** | | | |
| Fragility fracture | 26 (55.3%) | 15 (13.8%) | < 0.001 * |
| Hip fracture | 21 (44.7%) | 7 (6.4%) | < 0.001 * |
| Proximal humeral fracture | 1 (2.1%) | 3 (2.8%) | 1 |
| Pelvic fracture | 5 (10.6%) | 1 (0.9%) | < 0.01 * |
| Distal radius fracture | 0 (0%) | 4 (3.7%) | 0.32 |
| Spinal fracture | 3 (6.4%) | 8 (7.3%) | 1 |
| **Medication use** | | | |
| Zinc supplements | 0 (0%) | 0 (0%) | NA |
| Iron supplements | 5 (10.6%) | 2 (1.8%) | 0.03 * |
| Magnesium supplements | 10 (21.3%) | 15 (13.8%) | 0.24 |
| Vitamin D supplements | 4 (8.5%) | 16 (14.7%) | 0.43 |
| Calcium supplements | 3 (6.4%) | 8 (7.3%) | 1 |
| Proton pump inhibitors | 14 (29.8%) | 26 (23.9%) | 0.43 |
| H2 receptor antagonists | 0 (0%) | 8 (7.3%) | 0.11 |
| Diuretics | 8 (17.0%) | 14 (12.8%) | 0.62 |
| Antihypertensive | 31 (66.0%) | 56 (51.4%) | 0.11 |
| Opioid | 5 (10.6%) | 14 (12.8%) | 0.80 |
| Antibiotics | 2 (4.3%) | 2 (1.8%) | 0.58 |

* Statistically significant.

*(Continued)*

**Table 2.** (Continued)

** Participants were classified into three groups based on BMI according to the classification of the Japan Society for the Study of Obesity: High BMI (BMI ≥ 25), Normal (BMI 18.5 to 25), and Low BMI (BMI < 18.5).

BMI, body mass index; GERD, Gastroesophageal reflux disease.

**Table 3.** Laboratory Findings of patients with zinc deficiency and controls.

| Variables | Zinc deficiency (n = 47) | Control group (n = 109) | p value |
|---|---|---|---|
| Zinc, µg/dL | 46.2 ± 9.7 | 76.3 ± 16.5 | < 0.001 * |
| Corrected calcium ** | 9.6 ± 0.6 | 9.7 ± 0.5 | 0.492 |
| Hypocalcaemia (< 8.8 mg/dL) | 1 (2.1%) | 1 (0.9%) | 0.51 |
| Albumin, g/dL | 3.3 ± 0.7 | 4.0 ± 0.5 | < 0.001 * |
| Hypoalbuminaemia (< 3.8 g/dL) | 35 (74.5%) | 26 (23.9%) | < 0.001 * |
| Creatinine, mg/dL | 1.5 ± 1.8 | 1.4 ± 2.2 | 0.79 |
| eGFR, mL/min/1.73 m$^2$ | 60.8 ± 34.0 | 66.6 ± 28.3 | 0.28 |
| Haemoglobin, g/dL | 10.8 ± 2.0 | 13.1 ± 1.9 | < 0.001 * |
| Anaemia (Hb < 11.0 g/dL in Women, Hb < 13.0 g/dL in Men) | 33 (70.2%) | 18 (16.5%) | < 0.001 * |
| Haemoglobin A1c, % | 6.0 ± 1.0 | 5.9 ± 0.7 | 0.37 |
| CRP, mg/dL | 2.9 ± 4.2 | 1.2 ± 2.6 | < 0.01 * |
| ALP (IFCC), U/L | 89.9 ± 31.2 | 89.8 ± 38.7 | 0.99 |

* Statistically significant.

** The corrected calcium concentration was determined using the following formula. Corrected Ca concentration = measured Ca concentration + (4 - albumin concentration).

ALP, Alkaline phosphatase; CRP, C-reactive protein; eGFR, estimated glomerular filtration rate; IFCC, International Federation of Clinical Chemistry and Laboratory Medicine

## Discussion

This study demonstrated that anaemia, hip fracture, and hypoalbuminaemia were independent predictive factors of zinc deficiency in orthopaedic inpatients. Additionally, our study identified a correlation between spinal disease and a decreased likelihood of zinc deficiency.

Hip fracture was identified as one of the independent predictive factors of zinc deficiency in orthopaedic inpatients by our hospital-based study (OR, 6.24; 95% CI, 1.96–19.90; p < 0.01). Gau et al. conducted a retrospective prevalence study involving 157 community residents aged 50 years or older, regularly followed up at a geriatric clinic in southeast Ohio [28]. They concluded that a previous hip fracture was a significant predictive factor for zinc deficiency (OR, 9.65; 95% CI, 1.69–55.15; p = 0.011), consistent with the finding of our study. We speculate that hip fractures serve as a predictive factor of zinc deficiency because of the involvement of background factors such as osteoporosis, a well-established contributor to zinc deficiency [29]. A study involving 122 osteoporotic patients aged 65 years and over with zinc deficiency demonstrated a significant increase in bone mineral density from baseline after 6 and 12 months of zinc treatment [30]. Consequently, in cases where osteoporosis is present in patients with hip fractures, monitoring zinc levels is advisable, and zinc supplementation should be considered if a deficiency is detected.

Anaemia was also shown to be one of the predictive factors for zinc deficiency in orthopaedic inpatients in our multivariate logistic regression analysis (OR, 9.36; 95% CI, 3.35–26.10; p < 0.001). This result is consistent with that of Gau et al. (OR, 1.73; 95% CI, 0.60–4.95; p = 0.308) [28]. The association between zinc and anaemia has been extensively documented in women of reproductive age [31–34], preschool-aged children [35–37], and adults [38,39]. It is hypothesized that the mechanism underlying anaemia related to zinc deficiency involves impaired functionality of the zinc finger protein

**Table 4. Predictive factors associated with zinc deficiency.**

| Variables | Crude | | | Adjusted | | |
|---|---|---|---|---|---|---|
| | OR | 95% CI | *p* value | OR | 95% CI | *p* value |
| 50 and above | 3.87 | 0.85 - 17.60 | 0.079 | – | – | – |
| 60 and above | 4.67 | 1.55 - 14.10 | < 0.01 | – | – | – |
| 70 and above | 3.20 | 1.50 - 6.81 | < 0.01 | – | – | – |
| 80 and above | 4.76 | 2.20 - 10.30 | < 0.001 | – | – | – |
| 90 and above | 4.98 | 0.88 - 28.20 | 0.070 | – | – | – |
| Weight | 0.97 | 0.95 - 0.996 | 0.021 | – | – | – |
| High BMI | 0.35 | 0.15 - 0.83 | 0.017 | – | – | – |
| Low BMI | 2.15 | 0.92 - 5.04 | 0.079 | – | – | – |
| Fracture | 4.15 | 2.00 - 8.62 | < 0.001 | – | – | – |
| Spinal disease | 0.048 | 0.006 - 0.36 | < 0.01 | 0.056 | 0.006 - 0.49 | < 0.01 * |
| Osteoarthritis | 0.31 | 0.102 - 0.956 | 0.041 | | | |
| Fragility fracture | 7.76 | 3.51 - 17.10 | < 0.001 | | | |
| Hip fracture | 11.8 | 4.52 - 30.70 | < 0.001 | 6.24 | 1.96 - 19.90 | < 0.01 * |
| Pelvic fracture | 12.9 | 1.46 - 113.00 | 0.022 | – | – | – |
| Iron supplements | 6.37 | 1.19 - 34.10 | 0.031 | – | – | – |
| H2 receptor antagonists | < 0.001 | 0.000 - Inf | 0.99 | – | – | – |
| Antihypertensive | 1.83 | 0.90 - 3.73 | 0.094 | – | – | – |
| Hypoalbuminaemia | 9.31 | 4.23 - 20.50 | < 0.001 * | 3.63 | 1.33 - 9.89 | < 0.05 * |
| Anaemia | 11.9 | 5.33 - 26.60 | < 0.001 * | 9.36 | 3.35 - 26.1 | < 0.001 * |
| CRP | 1.18 | 1.04 - 1.34 | < 0.01 * | – | – | – |

\* Statistically significant.

BMI, body mass index; CI, confidence interval; CRP, C-reactive protein; OR, odds ratio

GATA-1, crucial for the differentiation and proliferation of erythroblasts [40]. Jeng and Chen conducted a literature review and concluded that zinc status should be carefully monitored and that zinc supplementation could prevent and treat anaemia [41]. If anaemia remains refractory to treatment, the possibility of zinc deficiency should be considered.

Hypoalbuminaemia was identified as one of the predictive factors for zinc deficiency in orthopaedic inpatients by this retrospective case-control study (OR, 3.63; 95% CI, 1.33–9.89; p < 0.05). This result is consistent with the following previous studies [25,28]. Gau et al. reported that hypoalbuminaemia (serum albumin < 3.5 g/dL) was risk factors associated with zinc deficiency (OR, 5.17; 95% CI, 1.80–14.90; p = 0.002) by the retrospective prevalence study took place from 2014 to 2017 [28]. Hennigar et al. also showed that study participants with hypoalbuminaemia (serum albumin ≤ 3.5 g/dL) were more likely to have low serum zinc concentrations (OR, 11.2; 99% CI, 3.4–37.3; p < 0.0001) [25]. Serum zinc concentrations may be influenced by the status of zinc-binding proteins, such as albumin, which binds approximately 80–85% of serum zinc [42]. Conversely, zinc deficiency may affect albumin synthesis. A prior study demonstrated that serum albumin concentrations improved following zinc supplementation in individuals with severe zinc deficiency [43]. Therefore, if either zinc deficiency or hypoalbuminaemia is identified, investigating the other is advisable.

Based on a national micronutrient survey, Gebremedhin reported that CRP concentration demonstrated statistically significant negative correlations with serum zinc in both children and women of reproductive age. The correlation coefficients between CRP and serum zinc were -0.159 in children (p < 0.001) and -0.019 in women (p = 0.010) [44]. Interestingly, our study yielded similar results, indicating that patients with zinc deficiency had significantly higher mean (SD) blood concentrations of CRP (2.9 mg/dL [4.2] vs. 1.2 mg/dL [2.6]; p < 0.01) than the control group. However, multivariate logistic regression analysis revealed that CRP concentration was not a predictive factor for zinc deficiency.

The strength of this study is that it investigated zinc deficiency in orthopaedic inpatients, which had not previously been investigated, and it identified predictors of zinc deficiency. On the other hand, this study has several limitations. First, it was a retrospective study with a relatively small sample size, and the data were confined to a single centre. Second, the study exclusively included Japanese individuals, and race or ethnicity was not considered as a variable in the logistic regression analysis. Third, background factors like osteoporosis were not assessed in multiple logistic regression analyses. Fourth, this study assessed zinc deficiency using solely serum zinc concentration as an indicator, excluding consideration of zinc intake and symptoms of zinc deficiency. This decision aligns with common practices in the field, where serum zinc levels are widely employed as a key marker for assessing zinc deficiency [22–25]. To overcome these limitations, it is imperative to conduct large-scale, multinational clinical trials. This study is anticipated to act as a stimulus for such initiatives.

## Conclusions

The study concluded that anaemia, hip fracture and hypoalbuminaemia were independent predictors of zinc deficiency in orthopaedic inpatients.

## Supporting information

**S1 Data.**
(XLSX)

## Author contributions

**Conceptualization:** Tomonori Shigemura.

**Data curation:** Daisuke Iida.

**Formal analysis:** Tomonori Shigemura.

**Investigation:** Tomonori Shigemura.

**Methodology:** Tomonori Shigemura.

**Project administration:** Yasuaki Murata.

**Resources:** Tomonori Shigemura.

**Software:** Tomonori Shigemura.

**Supervision:** Yasuaki Murata.

**Validation:** Daisuke Iida, Tomonori Shigemura, Yohei Yamamoto, Miki Morimoto.

**Visualization:** Yohei Yamamoto.

**Writing – original draft:** Tomonori Shigemura.

**Writing – review & editing:** Tomonori Shigemura.

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
