## [Decision Letter · Decision Letter 0]

18 Feb 2025

PONE-D-24-54135Hip fracture is one of the predictive factors of zinc deficiency in orthopaedic inpatients.PLOS ONE

Dear Dr. Shigemura,

Thank you for submitting your manuscript to PLOS ONE. After careful consideration, we feel that it has merit but does not fully meet PLOS ONE’s publication criteria as it currently stands. Therefore, we invite you to submit a revised version of the manuscript that addresses the points raised during the review process. If you are prepared to undertake the work required, I would be pleased to consider a revised version. 

For your guidance, reviewers' comments are appended below.

If you decide to revise the work, please submit a list of changes or a rebuttal against each point which is being raised when you submit the revised manuscript. 

We look forward to receiving your revised manuscript.

Kind regards,

Xinyan Bi

Academic Editor

PLOS ONE

Journal Requirements:

Reviewers' comments:

Reviewer's Responses to Questions

**Comments to the Author**

1. Is the manuscript technically sound, and do the data support the conclusions?

Reviewer #1: Partly

Reviewer #2: Yes

2. Has the statistical analysis been performed appropriately and rigorously? 

Reviewer #1: Yes

Reviewer #2: No

3. Have the authors made all data underlying the findings in their manuscript fully available?

Reviewer #1: No

Reviewer #2: Yes

4. Is the manuscript presented in an intelligible fashion and written in standard English?

Reviewer #1: Yes

Reviewer #2: Yes

5. Review Comments to the Author

Reviewer #1: Title:

I suggest changing the title to “Determinant of Zinc Deficiency in Orthopedic Inpatients” for clarity.

Abstract

Page 2, line 40: The study aimed to investigate the incidence of zinc deficiency. How did you assess the incidence with a case-control study design, given that you predetermined the number of case and control groups?

Introduction

The introduction is well-written, but please add details on the burden, consequences, and associated factors of the problem.

Page 3, line: Please clarify what you mean by retrospective case-control study design; it is unclear.

Materials and Methods

Page 4 line 129 “demographics (gender, age, height, weight, body mass index [BMI]),” How did you measure the weight and height of the patient. Which instrument was used?

If your focus is on incidence, which measures did you use: incidence density or cumulative incidence?

What analysis method did you employ to estimate incidence?

What effect measures did you use, and how was the analysis conducted?

Results

While you covered much content, you overlooked an essential component—interpretation of the predictive factors of zinc deficiency.

How did you control for confounding?

If you performed multivariate logistic regression analysis, please provide the values and regression table that includes both adjusted and crude odds ratios.

Discussion

If you used a case-control study design, why focus on the prevalence of the problem?

The discussion requires further synthesis and should concentrate more on the predictive factors.

What was the strength and the limitation of your study? Because it is the base for future research.

Reviewer #2: This study underlined hip fracture as one of the predictive factors of zinc deficiency in orthopaedic inpatients. The manuscript is quite well written. However, below is one concern on data analysis.

Independent variables of age (yr) and the categories (above 50, above 60, etc.) as well as BMI (kg/m2) and the categories (low BMI, etc.) are actually similar. Authors should choose which variables are included in the multivariate analysis to prevent having the same variables in one regression equation.

6. PLOS authors have the option to publish the peer review history of their article (what does this mean? ). If published, this will include your full peer review and any attached files.

**Do you want your identity to be public for this peer review?** For information about this choice, including consent withdrawal, please see our Privacy Policy .

Reviewer #1: No

Reviewer #2: No

---

## [Author Response · Author response to Decision Letter 1]

2 Mar 2025

Reviewers' comments:

Reviewer's Responses to Questions

Reviewer #1

Dear Reviewer,

We would like to express our great appreciation for the insightful comments. We have revised the manuscript according to the suggestions and believe that we have made considerable improvements.

If you think these changes are not enough, please let us know. We will follow your suggestions.

Title

I suggest changing the title to “Determinant of Zinc Deficiency in Orthopedic Inpatients” for clarity.

Response: Following the comment, we have changed the title to “Determinant of zinc deficiency in orthopaedic inpatients”.

Abstract

Page 2, line 40: The study aimed to investigate the incidence of zinc deficiency. How did you assess the incidence with a case-control study design, given that you predetermined the number of case and control groups?

Response: As you pointed out, it is not possible to study the incidence through case-control studies. We have therefore modified our study to focus on the predictors of zinc deficiency rather than the incidence. We have changed the abstract as follows.

“This study aimed to investigate predictive factors of zinc deficiency in orthopaedic inpatients.”

In addition, we have eliminated the parts of the incidence from all the parts of the manuscript.

Introduction

The introduction is well-written, but please add details on the burden, consequences, and associated factors of the problem.

Response: Following the comment, we have added the following text in the Introduction.

“…

Delayed wound healing after joint replacement surgery increases the risk of prosthetic joint infection (PJI) [11,12]. PJI is one of the most serious complications after arthroplasties, not only reducing patients' quality of life [13], but also leading to longer hospital stays [14] and increased healthcare costs [15]. A study using data from the Nationwide Inpatient Sample in the United States demonstrated that length of stay was significantly longer for infected hip (9.7 days) and knee (7.6 days) arthroplasties compared to uninfected procedures (hip, 4.3 days; knee, 3.9 days) (p < 0.0001). The study also showed that hospitalisation charges were significantly greater for infected arthroplasties than for uninfected arthroplasties (hips, 1.76 times; knees, 1.52 times) (p < 0.0001) [16].

…”

If you think this modification is insufficient, please let us know. We will follow your suggestion.

Page 3, line: Please clarify what you mean by retrospective case-control study design; it is unclear.

Response: To clear the meaning of the word, we have changed the text as follows.

“To address these concerns, we conducted a retrospective case-control study, including patients with zinc deficiency in the case group and those without zinc deficiency in the control group, to examine predictive factors of zinc deficiency in hospitalised patients.”

If you think this change is not enough, please let us know. We will follow your suggestion.

Materials and Methods

Page 4 line 129 “demographics (gender, age, height, weight, body mass index [BMI]),” How did you measure the weight and height of the patient. Which instrument was used?

Response: Based on the comment, we have added the following sentence in Materials and Methods.

“Height and weight were measured using height scales (Okada medical supply Co., Ltd.) and weight scales (TANITA Corporation), respectively.”

If your focus is on incidence, which measures did you use: incidence density or cumulative incidence?

Response: In accordance with the previous comment, we have modified our study to focus on the predictors of zinc deficiency rather than the incidence and removed the relevant parts of the incidence.

What analysis method did you employ to estimate incidence?

Response: In accordance with the previous comment, we have modified our study to focus on the predictors of zinc deficiency rather than the incidence and removed the relevant parts of the incidence.

What effect measures did you use, and how was the analysis conducted?

Response: Unfortunately, we could not understand what effect measures meant. We have excluded the relevant parts of the incidence according to the previous comment, is this an adequate response to this comment? If this comment has not been resolved, please provide the meaning of effect measures.

And by the analysis, do you mean statistical analysis methods? If so, we have already noted the following in Materials and Methods section. If you think this is insufficient, please let us know. We will follow your suggestion.

“Statistics

 ... To identify factors associated with zinc deficiency, multivariate logistic regression analysis (backward stepwise regression model) was performed for items that were significantly different from the univariate analysis. … All statistical analyses were performed using EZR version 1.52 [27], which is a graphical user interface for R version 4.02 (The R Foundation for Statistical Computing, Vienna, Austria).

 ...”

Results

While you covered much content, you overlooked an essential component—interpretation of the predictive factors of zinc deficiency.

Response: We have presented the results of predictive factors of zinc deficiency in the last section of the Results. For my previous experience, the results were interpreted in the Discussion section. Therefore, this time the interpretation of the results has been done in the Discussion section. If you feel that the interpretation should be moved to the Results section, please let us know. We will follow your suggestion.

How did you control for confounding?

Response: In accordance with the next comment, we have presented the crude and adjusted odds ratios for the variables in table 4. For variables that were significantly different in the univariate analysis, or for those that were not significantly different but had a p value of less than 0.15, we performed the multivariate logistic regression analysis (backward stepwise regression model) and allowed us to show the independent variables by excluding confounding. Therefore, the answer to this comment is the multivariate logistic regression analysis (backward stepwise regression model) was used to control for confounding. If you think this answer is not sufficient, please let us know. We will follow your suggestion.

If you performed multivariate logistic regression analysis, please provide the values and regression table that includes both adjusted and crude odds ratios.

Response: Following the comment, the crude odds ratios, 95% CI and p value have been added in Table4. If you think this change is not insufficient, please let us know. We will follow your suggestion.

Discussion

If you used a case-control study design, why focus on the prevalence of the problem?

Response: In accordance with the previous comment, we have modified our study to focus on the predictors of zinc deficiency rather than the incidence and removed the relevant parts of the incidence.

The discussion requires further synthesis and should concentrate more on the predictive factors.

Response: Following the comment, we have modified the Discussion section as follows.

“Discussion

This study demonstrated that anaemia, hip fracture, and hypoalbuminaemia were independent predictive factors of zinc deficiency in orthopaedic inpatients.

…

Hip fracture was identified as one of the independent predictive factors of zinc deficiency in orthopaedic inpatients by our hospital-based study (OR, 6.24; 95% CI, 1.96–19.90; p < 0.01).

…

Anaemia was also shown to be one of the predictive factors for zinc deficiency in orthopaedic inpatients in our multivariate logistic regression analysis (OR, 9.36; 95% CI, 3.35–26.10; p < 0.001). This result is consistent with that of Gau et al. (OR, 1.73; 95% CI, 0.60–4.95; p = 0.308) [28].

…

Hypoalbuminaemia was identified as one of the predictive factors for zinc deficiency in orthopaedic inpatients by this retrospective case-control study (OR, 3.63; 95% CI, 1.33–9.89; p < 0.05). This result is consistent with the following previous studies [25,28]. Gau et al. reported that hypoalbuminaemia (serum albumin < 3.5 g/dL) was risk factors associated with zinc deficiency (OR, 5.17; 95% CI, 1.80–14.90; p = 0.002) by the retrospective prevalence study took place from 2014 to 2017 [28]. Hennigar et al. also showed that study participants with hypoalbuminaemia (serum albumin ≤ 3.5 g/dL) were more likely to have low serum zinc concentrations (OR, 11.2; 99% CI, 3.4–37.3; p < 0.0001) [25].

…”

If you feel this modification is insufficient, please tell us. We will follow your suggestion.

What was the strength and the limitation of your study? Because it is the base for future research.

Response: Based on the comment, we have added the following text in the Disucussion.

“The strength of this study is that it investigated zinc deficiency in orthopaedic inpatients, which had not previously been investigated, and it identified predictors of zinc deficiency. On the other hand, this study has several limitations. …”

Reviewer #2

Dear Reviewer,

We would like to express our great appreciation for the insightful comments. We have revised the manuscript according to the suggestions and believe that we have made considerable improvements.

This study underlined hip fracture as one of the predictive factors of zinc deficiency in orthopaedic inpatients. The manuscript is quite well written. However, below is one concern on data analysis.

Independent variables of age (yr) and the categories (above 50, above 60, etc.) as well as BMI (kg/m2) and the categories (low BMI, etc.) are actually similar. Authors should choose which variables are included in the multivariate analysis to prevent having the same variables in one regression equation.

Response: Following the comment, age and BMI values were excluded from multivariate analysis. Albumin and haemoglobin values were also excluded from the analysis for the same reason.

---

## [Decision Letter · Decision Letter 1]

17 Mar 2025

Determinant of zinc deficiency in orthopaedic inpatients

PONE-D-24-54135R1

Dear Dr. Shigemura,

We’re pleased to inform you that your manuscript has been judged scientifically suitable for publication and will be formally accepted for publication once it meets all outstanding technical requirements.

Kind regards,

Xinyan Bi

Academic Editor

PLOS ONE

Additional Editor Comments (optional):

Reviewers' comments:

Reviewer's Responses to Questions

**Comments to the Author**

1. If the authors have adequately addressed your comments raised in a previous round of review and you feel that this manuscript is now acceptable for publication, you may indicate that here to bypass the “Comments to the Author” section, enter your conflict of interest statement in the “Confidential to Editor” section, and submit your "Accept" recommendation.

Reviewer #1: All comments have been addressed

Reviewer #2: All comments have been addressed

2. Is the manuscript technically sound, and do the data support the conclusions?

Reviewer #1: Partly

Reviewer #2: Yes

3. Has the statistical analysis been performed appropriately and rigorously? 

Reviewer #1: Yes

Reviewer #2: Yes

4. Have the authors made all data underlying the findings in their manuscript fully available?

Reviewer #1: Yes

Reviewer #2: Yes

5. Is the manuscript presented in an intelligible fashion and written in standard English?

Reviewer #1: Yes

Reviewer #2: Yes

6. Review Comments to the Author

Reviewer #1: (No Response)

Reviewer #2: (No Response)

7. PLOS authors have the option to publish the peer review history of their article (what does this mean? ). If published, this will include your full peer review and any attached files.

**Do you want your identity to be public for this peer review?** For information about this choice, including consent withdrawal, please see our Privacy Policy .

Reviewer #1: No

Reviewer #2: No

---

## [Editor Report · Acceptance letter]

PONE-D-24-54135R1

PLOS ONE

Dear Dr. Shigemura,

I'm pleased to inform you that your manuscript has been deemed suitable for publication in PLOS ONE. Congratulations! Your manuscript is now being handed over to our production team.

Kind regards,

on behalf of

Dr. Xinyan Bi

Academic Editor

PLOS ONE